# Your Body as a Tool to Learn Second Language Vocabulary

**DOI:** 10.3390/bs15080997

**Published:** 2025-07-22

**Authors:** Manuela Macedonia

**Affiliations:** Department of Information Engineering, Johannes Kepler University, 4040 Linz, Austria; manuela.macedonia@jku.at

**Keywords:** embodiment, second language learning, memory, learning, instruction, education, language

## Abstract

Vocabulary acquisition is a fundamental challenge in second language (L2) learning. Recent research highlights the benefits of using gestures to enhance vocabulary retention. This comprehensive review explores the theoretical, empirical, and neuroscientific foundations of gesture-enhanced learning. Findings show that the human body, specifically sensorimotor engagement, can be harnessed as an effective cognitive tool to support long-term word learning. This paper examines the limitations of traditional vocabulary learning methods, introduces embodied cognition as a theoretical framework, presents behavioral and neuroscientific evidence supporting gesture-based learning, and offers practical applications for educational settings. This integration of multidisciplinary research provides a robust foundation for reconceptualizing the role of physical engagement in second language acquisition.

## 1. Introduction

Vocabulary acquisition remains one of the most persistent and complex challenges for second language (L2) learners. The scale of the vocabulary required for functional language proficiency underscores the magnitude of the challenge faced by L2 learners. Research by [73] ([73]) estimates that individuals need to understand approximately 8000 to 9000 words to read and comprehend authentic texts, such as newspapers or novels, with minimal difficulty. This high lexical threshold represents a considerable cognitive burden, particularly when traditional techniques like word list memorization are used as the primary method of learning.

Traditional approaches to vocabulary learning have revealed significant limitations that hinder optimal acquisition. [38] ([38]) highlights the problem of decontextualized learning, in which vocabulary items are presented in isolation, lacking the meaningful contexts that support deeper cognitive processing and long-term retention. This approach fails to situate words within the rich semantic and pragmatic environments where they naturally occur. Compounding this issue, [100] ([100]) observes that conventional methods often position learners as passive recipients rather than active meaning-makers, resulting in superficial engagement with the target vocabulary. The learner becomes a container to be filled with information rather than an agent constructing meaningful linguistic connections. Furthermore, [49] ([49]) point out that traditional approaches typically engage only visual and occasionally auditory modalities. Traditional approaches thus neglect the potential benefits of multimodal learning that would create stronger memory traces through multiple sensory pathways. This restricted sensory engagement contrasts sharply with how we naturally learn language through multisensory experiences. Perhaps most fundamentally, [5] ([5]) identifies a cognitive–linguistic bias in traditional vocabulary instruction, where the predominant focus on mental processes overlooks the potential role of the body in language learning.

However, vocabulary knowledge is not a simple matter of knowing word meanings. As [92] ([92]) emphasizes, vocabulary competence is multidimensional. It encompasses not only the ability to associate word forms with their meanings but also includes knowledge of word collocations (i.e., which words tend to appear together), grammatical behavior (e.g., verb patterns), register, and sociolinguistic appropriateness. This layered nature of vocabulary requires instructional approaches that can address multiple aspects of language use simultaneously, a challenge that simplistic memorization techniques are ill equipped to meet.

Among the strategies employed over the decades, rote learning, based on repetition and memorization has been especially prevalent ([112]). One common method within this approach is the use of bilingual word lists, which allow learners to rapidly accumulate vocabulary. However, while such methods may provide quick initial exposure to new terms, their long-term effectiveness has been questioned ([3]). Studies suggest that rote learning yields weaker retention outcomes when compared to more elaborative techniques that engage learners in deeper cognitive processing ([61]).

Alternative strategies that promote meaningful associations and contextual understanding include methods where learners connect new vocabulary to prior knowledge or use it within contextually rich environments, both of which facilitate memory encoding and retrieval. Additionally, mnemonic devices and visual imagery have been shown to support memorization and recall by engaging mental visualization and associative networks ([2]).

In recent years, digital technology has added a new dimension to vocabulary learning. Language learning apps, online platforms, and gamified environments now offer learners interactive experiences that go beyond static memorization ([53]). These tools often include pedagogically informed features such as spaced repetition algorithms and real-time feedback, which are known to enhance retention and engagement. Moreover, technology enables learners to engage in self-regulated learning (SRL), where they take responsibility for managing their own learning processes ([12]). SRL is especially crucial in language acquisition, as classroom instruction alone typically does not provide the extensive exposure and practice necessary to develop a robust lexicon ([102]).

Despite the diversity of available methods and technologies, research in the field remains fragmented. Many studies differ in terms of design, scope, and assessment metrics, making it difficult to directly compare their outcomes or synthesize consistent conclusions. Furthermore, although these methods vary in form, they often share a common characteristic insofar as vocabulary learning is usually conducted in sedentary settings, i.e., reading, listening, observing, or interacting with screens. As such, the sensory and physical dimensions of learning are largely underutilized, raising questions about whether more embodied approaches might better address the complex demands of vocabulary acquisition.

Altogether, numerous methods and countless studies exhibit varying focuses, experimental designs, and outcomes that are not directly comparable. While each method may be effective for specific aspects of vocabulary learning, their overall efficacy has not yet been systematically evaluated, neither through behavioral studies nor from a neuroscientific perspective. Furthermore, all these methods share the common characteristic that the learning process is limited to reading, listening, watching, observing, and repeating, typically conducted while sitting.

## 2. Why the Body Matters: Embodied Cognition

Is the mind separated from the body? René Descartes in his *Discours de la Methode* ([19]) posited that the mind and body are two distinct substances; the mind was conceptualized as a non-material, thinking entity (res cogitans) and the body as a material, non-thinking entity (res extensa). In those days, his dualistic perspective aimed to reconcile the mechanistic view of the physical world with the existence of consciousness and rational thought.

Descartes’ approach became influential in Western philosophy, science, and most importantly for us, education. On this basis, educational paradigms have promoted a separation between mental and physical processes in the last centuries. It is no surprise that even today, abstract and symbolic reasoning (e.g., reading, writing, calculation, learning) continue to be privileged over bodily experience, movement, and emotion in educational settings. The Cartesian dualist view, which separates mind and body, stands in contrast to a large body of empirical evidence suggesting that cognition is fundamentally grounded in sensorimotor processes. In other words, traditional education mostly reflects dualistic assumptions by minimizing physical engagement. Students are typically required to sit still, suppress bodily movement, and focus on purely cognitive tasks, although the literature has long suggested that restricting physical movement in learning environments may hinder memory consolidation, conceptual understanding, and learner engagement ([110]).

In her paper, Wilson articulates six key claims that collectively argue for an embodied view of cognition. First, cognition is situated, that is, it takes place in the context of a real-time environment and is shaped by the physical and social settings in which it occurs. Rather than being generalized problem-solving detached from the immediate surroundings, thinking is often intimately tied to the specific situation at hand. Second, cognition is time-pressured. In everyday tasks, individuals rarely have the luxury of prolonged contemplation; instead, they must act and decide quickly, often relying on sensorimotor cues to guide behavior. Third, Wilson argues that we off-load cognitive work onto the environment. This means that rather than carrying all necessary information internally, humans often structure their physical surroundings in ways that reduce cognitive load, such as writing reminders, using gestures, or arranging objects spatially to support memory and planning. Fourth, the environment is part of the cognitive system itself. According to this claim, cognition does not reside solely in the brain but emerges from dynamic interactions between the mind, body, and world. Fifth, cognition is for action. This point shifts the purpose of thought from abstract reasoning to practical, goal-directed behavior. Our mental processes have evolved not for passive contemplation but to support adaptive, real-time interactions with the environment. Finally, Wilson emphasizes that offline cognition is body-based, even when people are not directly interacting with the world, their mental simulations of objects, actions, or scenarios are grounded in sensorimotor experiences. For example, imagining grasping an object activates similar neural systems as actually performing the grasp ([87]).

These claims refute the notion that cognition is an abstract, internal computation disconnected from the body and the world and underscore the embodied nature of cognition. They challenge classical cognitive models that treat the mind as a disembodied information processor and instead propose that cognitive processes are best understood as emerging from real-world, bodily engagements. This reconceptualization has profound implications for fields like language learning, where traditional approaches often disregard the role of movement, gesture, and spatial interaction. By recognizing that sensorimotor experiences are not peripheral but central to thought, Wilson’s framework supports a shift toward more embodied, interactive methods of instruction.

Contrary to the traditional view that cognition relies on amodal, abstract symbols detached from sensory experience, [7] ([7]) introduces the concept of grounded cognition. He proposed that cognitive representations and processes, including language, are fundamentally grounded in perceptual and sensorimomotor experiences. His theory posits that understanding concepts involves simulating sensory experiences, thereby linking cognition directly to bodily states and actions. The role of the living body in grounding cognition is also emphasized by [21] ([21]) in the enactive approach. They argue that homeostatic and allostatic processes are fundamental to shaping sensorimotor interactions and, consequently, cognitive processes. This view underscores the importance of the body’s internal regulatory mechanisms in the development and functioning of cognition.

## 3. Embodied Language

Following this line of reasoning, language, being a cognitive process, must also be embodied. This contradicts the position sustained by [13] ([13]) and [28] ([28]) that language is a module of the mind. The modularity hypothesis is an idea that was central in Chomsky’s *Universal Grammar*. It proposed that humans are born with a language-specific faculty, the language acquisition device (LAD) that “emerges” during childhood. This hypothesis had and still has a strong influence on how foreign language is understood, learned, and taught as a “phenomenon of the mind”.

In the 1990s, Elizabeth Bates and Jeffrey Elman developed theoretical models known as connectionist and emergentist approaches, which challenged the idea that the human brain contains a specialized, built-in “language module”. Instead of assuming that language is processed by a distinct, isolated part of the mind, they proposed that language abilities emerge gradually from general learning mechanisms, such as pattern recognition, memory, and statistical learning, that are used across many different cognitive tasks. Their research shows how complex language structures can be learned through repeated exposure and experience, without requiring an innate, hardwired system dedicated solely to language. Their work demonstrates that language learning can be explained by domain-general learning mechanisms operating over richly structured input. Through neural network simulations, they showed that complex syntactic patterns can emerge without invoking innate grammatical knowledge ([25]). They further emphasized that language processing is distributed across the brain and intertwined with general cognitive and perceptual systems, contradicting claims of strict modularity. Together, their work supports a model of language acquisition grounded in interaction, plasticity, and competition between forms ([8]).

Barsalou’s theory of perceptual symbol systems ([6]) marked a significant shift in how cognitive scientists conceptualize mental representation. Contrary to the traditional view that cognition operates on amodal, abstract symbols divorced from sensory experience, Barsalou proposed that cognitive processes, including language, are fundamentally grounded in perceptual and sensorimotor experiences. According to this theory, mental representations are not encoded in a purely symbolic or propositional format but are instead formed from reactivations of sensory and motor states that occurred during initial experiences. These reactivations, or simulations, enable individuals to “re-experience” perceptual components of previous encounters, which in turn support reasoning, memory, and language comprehension.

Subsequent empirical research across fields such as neuroscience, psycholinguistics, and developmental psychology has consistently reinforced this view. Functional imaging studies demonstrate that language comprehension activates modality-specific brain regions associated with action, vision, and sensation ([7]; [84]). For instance, when participants read or hear action-related words like “kick” or “grasp”, motor areas of the brain involved in leg or hand movements become active, even in the absence of any physical movement ([33]). Behavioral research further supports this connection between language and action: the action–sentence compatibility effect (ACE) shows that comprehending sentences involving motion affects motor responses, indicating that language processing is dynamically linked to sensorimotor systems ([32]). These findings suggest that language processing is not confined to abstract, symbolic computations but is instead distributed across the same neural and cognitive systems involved in perception and action ([27]; [29]; [114]).

The framework of embodied cognition builds on these insights. It argues that language acquisition, processing, and use are inseparably linked to the body’s sensorimotor systems. From this perspective, understanding language is not a matter of decoding arbitrary symbols; rather, it involves mentally simulating the physical experiences those words represent. This simulation allows the brain to reconstruct the sensory, motor, and emotional aspects of meaning, thereby grounding linguistic expressions in the lived experience of the body. In other words, language is not just about the world, it is experientially tied to how we act in, perceive, and navigate that world.

As [82] ([82]) emphasizes, this view challenges long-held assumptions about the disembodied nature of linguistic knowledge. Language is not learned or processed in a vacuum but emerges through continuous interaction between the body, the mind, and the environment. This has profound implications for language and word learning, and pedagogy, suggesting that approaches which incorporate movement, gesture, and sensory engagement may align more closely with the natural architecture of the human cognitive system.

The embodied language perspective becomes evident when simply observing how children acquire their first language. From the earliest stages of development, infants engage with language not just passively through listening but actively through multimodal, embodied interactions with their caregivers and their environment. Infants reach, point, mimic facial expressions, grasp objects, and perform gestures, all while caregivers speak and respond. These bodily actions are not separate from language learning; rather, they form the very foundation upon which linguistic meaning is built. Tomasello’s extensive empirical work demonstrates, as synthesized in his book Constructing a Language ([103]), that children acquire their native language by building it up through experiential, interaction-based learning.

Collectively, these studies underscore the central tenet of embodied cognition, i.e., the body is not merely a vessel for the brain but an active participant in the formation and operation of cognitive processes. This perspective has profound implications for various fields, including education, where acknowledging the embodied nature of cognition can lead to more effective strategies and technologies.

## 4. Gesture as a Bridge Between Body and Language

Gestures hold a unique and powerful position at the crossroads of language, cognition, and bodily experience. Far from being peripheral to communication, gestures are now widely recognized as an integral component of the language system. Pioneering work by [70] ([70]) proposed that gesture and speech form a single, unified system, working together to convey meaning. While speech typically conveys information in a segmented, linear, and analytic fashion, gestures provide imagistic, holistic, and synthetic representations. This complementary relationship enhances the communicative richness of spoken language and offers a promising mechanism for supporting vocabulary acquisition, particularly in second language (L2) learning.

[43] ([43]) provided a comprehensive framework for understanding gesture as an integral component of human communication. Rather than treating gesture as a supplement to language, Kendon positions it as a form of “utterance” in its own right, embedded in social interaction and governed by semiotic principles. He argued that gestures are visible actions that convey meaning through their formal features, timing, and relation to speech. Kendon also explored how gesture spans a continuum from spontaneous, idiosyncratic movements to highly conventionalized forms like emblems. This perspective supports the view that gesture is a key modality in meaning-making, allowing speakers to “fabricate” meaning dynamically with the body, especially the hands. His work challenges educators and linguists alike to recognize the role of bodily action not merely as expressive ornamentation but as a central, semiotic process in communication and learning.

More recently, [98] ([98]) emphasizes that gestures emerge from practical, hands-on actions and evolve into socially shared symbolic forms through repetition—what he terms “conceptual gestures”. They enable displaced reference and conceptual representation in communicative interaction. Streeck situates gesture within situated, multimodal interaction, intertwined with speech, gaze, and posture, rather than as an independent code. In earlier work, [97] ([97]) similarly characterizes gesture as a “craft”, an improvised, context-sensitive art rooted in bodily skill, not a fixed symbolic system.

One key mechanism by which gestures support learning is gesture-speech synchrony. When gestures occur in temporal alignment with speech, they create integrated moments of multimodal coherence, allowing learners to link verbal input with visual and motor information. This temporal synchronization enhances encoding by reinforcing connections across sensory modalities ([24]). Studies show that such synchronized input fosters better memory consolidation ([74]), particularly when learning unfamiliar or abstract terms.

Another influential mechanism is representational resonance, particularly through the use of iconic gestures, those that visually resemble the meaning of a word. Iconic gestures provide semantic redundancy, effectively representing the same concept through both linguistic and visual channels. This redundancy strengthens semantic processing and aids in conceptual grounding ([83]), especially when learners are dealing with novel lexical items.

In addition to visual and semantic effects, gestures also engage the motor system, creating what researchers refer to as motor resonance. When learners observe or produce gestures during language learning, the associated motor representations become linked to the corresponding linguistic forms ([61]; [88]). This motor activation facilitates later recall, as it provides additional sensorimotor pathways through which the word can be retrieved ([36]). This process is closely tied to embodied cognition theories, which suggest that bodily states and actions play a fundamental role in mental representations.

A fourth mechanism, often referred to as cross-modal binding, involves the integration of linguistic and motor experiences into a single memory trace ([109]). The combination of auditory (speech), visual (gesture), and motor (movement) cues strengthens the associative network surrounding a word, making it easier to recall through multiple retrieval routes ([71]). This rich encoding is particularly beneficial in the context of vocabulary learning, where learners must not only memorize a form-meaning connection but also activate it in context ([68]).

Empirical evidence from second language acquisition supports these theoretical claims. Research shows that pairing new vocabulary with meaningful, congruent gestures leads to enhanced retention and retrieval, especially when the gestures are actively produced by learners rather than passively observed. Such multimodal learning episodes engage a variety of cognitive and sensory systems, including vision, hearing, kinesthetic feedback, and affective engagement ([54]).

These multisensory imprints create durable memory traces ([113]) and turn the body into an instrument of learning. In this way, gesture serves as a bridge, not merely between body and language, but also between sensory experience and abstract knowledge ([55]), offering a pedagogically valuable tool in vocabulary acquisition.

In a recent paper, [46] ([46]) argues for a fundamental rethinking of second language pedagogy by emphasizing the role of the body in the learning process. He contends that traditional approaches often ignore the embodied and multimodal nature of communication, which includes not only speech but also gesture, posture, facial expression, and spatial interaction. Drawing on the work of McNeill, Kendon, and Streeck, Lapaire positions gesture as an essential part of meaning-making that is integral to speech rather than an optional supplement. Lapaire shows how gestures can facilitate comprehension, enhance memory, and support expressive capacity, especially when teaching abstract or complex language structures. The paper draws on principles from embodied cognition and cognitive linguistics, arguing that language is grounded in physical experience and shaped by the body’s interaction with the world. By engaging learners in physical activities like miming, gesturing, and spatial mapping, instructors can help internalize linguistic patterns in meaningful, memorable ways. Although such methods may initially seem unconventional, Lapaire makes a compelling case that embracing expressive, body-based teaching enriches the classroom experience and aligns with how humans naturally communicate.

## 5. Types of Gesture and Their Differential Effects

While the general effectiveness of gestures in supporting second language vocabulary learning is well established, not all types of gesture contribute equally to this process. Differences in gestures’ form, semantic alignment, and cognitive load can significantly influence their pedagogical value. Among the most extensively studied are iconic gestures, which visually represent aspects of a word’s meaning. For instance, mimicking the motion of drinking to accompany the verb to drink directly maps physical action onto the lexical item. Studies such as those by [55] ([55]) have found that iconic gestures are particularly effective in enhancing the acquisition of concrete nouns. These gestures help create a multimodal representation of meaning that facilitates deeper semantic processing and better long-term retention.

For more abstract vocabulary, metaphoric gestures offer a useful alternative. Rather than mimicking a literal action, these gestures convey abstract ideas through metaphorical representation, for example, raising the hands upward to indicate the concept of increase ([55]). Research by [37] ([37]) suggests that such gestures can be instrumental in supporting comprehension and recall of less tangible lexical items, though they may require more instructional scaffolding due to the lack of an immediate sensory–motor link.

In the domain of spatial language and function words, deictic gestures, i.e., gestures that involve pointing, have demonstrated particular utility. By physically indicating locations, directions, or referents, deictic gestures help learners raise attention anchor linguistic expressions in the immediate environment ([11]). [111] ([111]) provide evidence that such gestures facilitate learning of prepositions and other spatial terms by creating a clear connection between language and spatial cognition.

Another class of gestures, beat gestures, consists of rhythmic, non-semantic movements that often accompany speech to mark emphasis or prosodic rhythm. Although beat gestures do not convey lexical meaning directly, they may still support vocabulary learning by enhancing auditory salience and helping to segment input. However, research by [96] ([96]) indicates that the effects of beat gestures are typically more modest compared to those of iconic or metaphoric gestures, particularly when it comes to semantic recall.

Perhaps the most immersive form of gestural involvement comes from enactments or full-body pantomimes. These involve learners physically acting out the meaning of a word, often incorporating whole-body movement. [4] ([4]) total physical response method was among the earliest to advocate for this approach, and subsequent research by [104] ([104]) has confirmed that enactments yield robust benefits, especially for action-related vocabulary.

Underlying the effectiveness of these gesture types is the principle of semantic congruency. As [40] ([40]) shave shown, gestures that are semantically aligned with the word they accompany significantly enhance learning outcomes, while incongruent or arbitrary movements may confuse learners or impose additional cognitive load. Congruency ensures that the gesture serves as a meaningful cue, reinforcing rather than competing with the verbal input. This suggests that gesture-based instruction should prioritize meaningful, representative movements that map clearly onto the lexical content being taught ([48]).

In summary, the type of gesture used in vocabulary instruction plays a critical role in determining learning outcomes. Iconic and enactment gestures tend to offer the strongest benefits, particularly when they align closely with the semantics of the target word. Metaphoric and deictic gestures also offer valuable support, especially for abstract and spatial language, respectively. Even beat gestures, while less impactful in terms of meaning, may still contribute to prosodic awareness and learner engagement. Ultimately, the pedagogical effectiveness of any gesture hinges on its congruency with the target vocabulary and its ability to engage learners in multimodal processing.

## 6. Learning by Doing: Behavioral Evidence

Over the past decades, a growing body of research has explored the impact of gestures on second language vocabulary acquisition. This interest stems from foundational cognitive theories such as Paivio’s Dual Coding Theory ([78]; [79], [80]), which posits that information encoded through both verbal and non-verbal channels, like gestures, is more easily retained and retrieved. Building on this premise, early work in psychology, notably by [26] ([26]) and [15] ([15]), established the “enactment effect”, demonstrating that physically performing actions associated with verbal expressions significantly enhances memory compared to verbal learning alone. These studies laid the groundwork for understanding the potential of embodied learning in language education.

In the context of second language instruction, the value of gestures became more explicitly studied in the 1990s. [1] ([1]) investigated the use of emblematic gestures in the teaching of French idiomatic expressions, finding that gestures helped learners form and access mental representations of new vocabulary more effectively. In her doctoral dissertation, [51] ([51]) designed a classroom-based experiment where university students were introduced to an artificial corpus of 36 novel words. One group of students learned vocabulary audio-visually (seeing and hearing the words), while another group learned the same words but additionally performed related gestures as they encoded. Students who learned with gestures consistently outperformed their peers in both immediate recall tests and long-term retention assessments conducted weeks and months later. This superior performance was interpreted as evidence that gestures facilitate deeper encoding of verbal material, probably by engaging multiple neural systems, including motor areas of the brain, in the learning process. These findings were published in English in a paper by [54] ([54]), supporting the theoretical framework of embodied cognition by demonstrating that bodily movement can enhance cognitive outcomes such as vocabulary learning. Moreover, that study emphasized the practical implications for second language instruction, advocating for the integration of gestures in teaching strategies to improve long-term retention. Macedonia and Klimesch concluded that the addition of gestures creates a richer and more robust memory trace than audio-visual input alone.

[99] ([99]) explored the role of teacher gestures in adult L2 classrooms, finding that learners benefited from enhanced listening comprehension and vocabulary recognition when instruction was accompanied by gestures and facial expression. [101] ([101]) showed that children exposed to gestures while learning foreign words not only remembered them better but also understood their meanings more deeply.

In their 2011 study, Macedonia et al. investigated the effects of iconic gestures on foreign language vocabulary acquisition using Vimmi an artificial language paradigm. Participants learned 92 pseudowords over four consecutive days. In the experimental condition, each word was paired with a consistent, semantically congruent iconic gesture, which participants performed during learning. In the control condition, participants also performed gestures, but these were semantically unrelated and randomly changed on each presentation, effectively preventing any meaningful association between gesture and word. Results on vocabulary retention showed that participants in the iconic gesture condition recalled significantly more words than those in the control condition with unrelated gestures. This confirmed that the semantic congruency between gesture and word meaning plays a crucial role in supporting vocabulary learning, beyond the effect of motor activity alone. Crucially, a follow-up test conducted two months later revealed that the advantage of the iconic gesture group persisted over time. Participants who had learned vocabulary with meaningful gestures continued to outperform the control group, demonstrating superior long-term retention. These results underscore the durability of memory traces formed through semantically rich, embodied learning. The study thus provided strong behavioral evidence that iconic gestures support both immediate and delayed recall, reinforcing the idea that bodily actions tied to word meaning serve as effective anchors in memory consolidation.

[55] ([55]) explored how different types of gestures influence vocabulary acquisition and use in a foreign language learning context. Participants were taught 32 sentences in an artificial language over six days, using a method that integrated gesture with spoken and written input. The sentences included both concrete content words (e.g., nouns, verbs) and abstract or function words (e.g., conjunctions, prepositions). The gestures accompanying the words were differentiated; for concrete words, learners performed semantically related iconic gestures, movements that visually resembled the meaning of the word. For abstract or grammatical words, the gestures were arbitrary but consistent, designed to create a motor code even though they bore no natural relationship to the words’ meanings. This design allowed the authors to examine how both types of gestures influenced language learning and use.

Behavioral results showed that words learned with gesture support, regardless of whether the gestures were iconic or arbitrary, were better retained than words learned without gesture. In a delayed transfer task, participants were asked to construct novel sentences using the vocabulary they had learned. Words that had been encoded with gestures were recalled more often and used more accurately, particularly those that had been paired with semantically meaningful gestures. The results suggest that iconic gestures enhance the memorability of concrete words, while arbitrary gestures still provide a motoric encoding advantage for abstract or functional items.

The role of gestures in facilitating long-term retention has also been further supported in a study by [69] ([69]). In their study, adult learners who physically performed semantically meaningful gestures during the learning phase were compared to peers who learned the same words through verbal presentation alone, or in conjunction with static visual imagery. The critical measure of interest was not just immediate learning outcomes, but the durability of word retention over time. The behavioral data clearly demonstrated a significant advantage for the gesture-enriched group. These participants showed markedly better recall and recognition of vocabulary, both immediately after learning and in delayed post-tests conducted days later. Importantly, the benefits of gesture use were not limited to immediate recall, but extended into the long term, underscoring the robustness of embodied memory traces. Mayer and colleagues concluded that gestures act as effective cognitive tools that not only aid comprehension during initial learning but also significantly improve the longevity of vocabulary retention. Their findings provide strong empirical support for integrating motor activity into language instruction and further establish gesture-based learning as a practical and impactful method within both experimental and classroom contexts.

More recently, [64] ([64]) and [65] ([65]) provided robust behavioral evidence for the effectiveness of gesture-enriched learning in second language acquisition. In their 2020 study, adult participants were taught foreign language vocabulary under different conditions; one group learned words using traditional audio-visual input, while the other group learned the same words accompanied by congruent gestures. The behavioral results revealed that participants in the gesture group significantly outperformed those in the non-gesture group in vocabulary recall tasks. Learners who used gestures were more accurate and faster in retrieving the learned words, even after a delay, indicating improved retention and access to lexical knowledge. The 2021 study extended these findings to younger learners by examining 12- and 14-year-old schoolchildren. Both age groups participated in vocabulary training sessions using either gesture-enriched or standard multisensory methods. Behaviorally, children in the gesture-based condition demonstrated superior vocabulary recall compared to their peers in the control group. Notably, the 14-year-olds showed a greater benefit from the gesture-based training than the younger group, suggesting that the effectiveness of sensorimotor-enriched learning strategies may increase with developmental maturation.

Further evidence comes from the work of Christian Andrä and colleagues ([2]), who examined the effects of gesture and picture enrichment in vocabulary learning among eight-year-old children. In this classroom-based study, children learned English vocabulary over five days using either gesture-enriched methods, picture-enriched methods, or standard auditory instruction. When tested at intervals of three days, two months, and six months post-training, both gesture and picture-enriched groups showed significantly better vocabulary retention than the control group. Interestingly, in contrast to findings with older learners, gestures did not outperform pictures in this younger age group; both modalities contributed equally to improved learning outcomes. These results suggest that while gesture-based learning is effective across age groups, its relative advantage may develop with age.

Recent evidence from [31] ([31]) further substantiates the role of gestures as powerful scaffolding tools in foreign language vocabulary acquisition. In a series of experiments, participants were taught new foreign vocabulary through multiple gesture-based instructional conditions. The researchers found that when gestures were semantically congruent with the words being learned, meaning the physical movement clearly mirrored or symbolized the word’s meaning, participants demonstrated significantly better retention and recall than under conditions where the gestures were incongruent or entirely meaningless. These results underscore the importance of gesture semantics; gestures that align with a word’s meaning appear to enhance the depth of encoding, making the words more memorable. In fact, brain imaging research by [60] ([60]) revealed increased activity in frontal brain regions associated with conflict and cognitive control when, in the scanner, participants were presented words that had been learned with incongruent gestures. This illustrates the cognitive cost of mismatched gesture–word pairings.

However, the study by García-Gámez and Macizo did not stop at semantic congruence. They also explored whether it mattered whether learners simply observed gestures or physically enacted them. Their findings revealed that active performance of gestures further strengthened vocabulary learning, particularly in conditions where semantic congruence was disrupted. While observing congruent gestures was generally effective, physically performing the gestures appeared to help learners resist the negative effects of incongruent pairings, likely due to deeper sensorimotor involvement. Taken together, the study by García-Gámez and Macizo offers strong behavioral evidence that gestures can significantly enhance vocabulary learning, but only when implemented meaningfully. The combination of semantic congruence and active participation appears to be essential for maximizing the instructional impact of gestures. Their findings have important implications for language pedagogy, suggesting that gestures should not be used indiscriminately, but rather designed with semantic intent and ideally performed by learners themselves to achieve optimal outcomes.

In two experiments by [91] ([91]), participants were presented with nouns and action verbs preceded by either iconic gestures, pictures, or no prime. Experiment 1 assessed free recall, while Experiment 2 evaluated recognition memory. The results indicated that in the free recall task, there were no significant differences among the three conditions. However, in the recognition task, words preceded by iconic gestures were recognized more accurately and quickly than those preceded by pictures or no prime. This advantage was more pronounced for verbs than for nouns.

With the advent of digital technology, researchers have begun exploring how virtual environments might incorporate gesture-based learning ([53]). In their study, [9] ([9]) explored how gesture-based learning could be integrated into digital environments by testing the effectiveness of a virtual agent capable of performing iconic gestures to teach foreign vocabulary. Participants were trained on a set of 36 words under three instructional conditions: one where a human trainer performed semantically meaningful gestures while introducing the words, one where a digital avatar named Billie enacted the same gestures, and a control condition that presented words audio-visually but without gesture support. Training took place over three consecutive days, and vocabulary retention was assessed immediately following training as well as four weeks later, to evaluate long-term effects. The behavioral results demonstrated that participants who learned with gesture support, whether from the human or virtual trainer, showed significantly better word recall than those in the control condition. These learners retained more words over time, suggesting that the integration of gestures, even in digitally mediated formats, enhances memory encoding and long-term retention. Bergmann and Macedonia concluded that gestures, even when delivered through virtual agents, can produce learning outcomes comparable to those achieved with human gesture instruction. This finding provides important support for the inclusion of embodied learning principles in computer-assisted language learning and points to the broader potential of gesture-enabled digital environments in education.

In another study, [90] ([90]) investigated how imitating iconic gestures performed by a virtual pedagogical agent and viewing pictures-could support vocabulary acquisition in digital learning environments. Participants learned foreign words representing nouns, verbs, and adjectives under three conditions: gesture imitation, picture viewing, and a control with no enrichment. The results showed that gesture imitation was particularly effective for learning nouns, while pictures were most beneficial for verbs. No significant benefit was observed for adjectives with either form of enrichment. The study concluded that the effectiveness of non-verbal instructional strategies depends on both the type of enrichment and the grammatical category of the vocabulary, underscoring the importance of aligning instructional methods with linguistic material in technology-enhanced language learning.

Much of the existing literature on embodied language learning has focused on (iconic) gestures, i.e., bodily movements that represent word meanings in symbolic or metaphorical ways. These gestures have been shown to enhance vocabulary learning by engaging the motor system and reinforcing semantic memory through physical enactment. Macedonia and colleagues ([56]) extended the gesture paradigm by investigating how physically interacting with virtual objects, particularly through grasping movements, affected vocabulary acquisition in a second language. This approach reflects developmental insights from early language learning, where children naturally explore and internalize meaning through tactile interaction, i.e., grasping, manipulating, and physically engaging with objects in their environment. These embodied actions are not peripheral but foundational to how meaning is constructed and language is learned. 

In that study, participants learned words from an artificial language within a virtual reality environment, where they were not only exposed to the auditory and visual forms of each word but also physically engaged by grasping corresponding virtual objects projected tridimensionally within the VR setting. Learners who interacted with the virtual objects through grasping showed significantly improved recall and faster word recognition compared to those who learned through auditory–visual input alone, or who only observed the virtual objects without physically engaging with them. These findings suggest that the tactile and motoric feedback involved in grasping supports deeper encoding of word-object associations, creating more robust memory traces than gesture alone.

Building on these insights, follow-up research by Macedonia and colleagues ([57]) further explored whether the benefits of grasping varied based on individual learner characteristics. This study revealed that grasping-based learning disproportionately benefited learners with lower language aptitude, who often struggled with traditional forms of instruction. For these learners, physically grasping and objects and thereby learning words offered a compensatory advantage, helping to anchor verbal information in embodied, sensorimotor experiences. In contrast, high-aptitude learners did not show significant gains from grasping, suggesting that the additional cognitive support provided by the motor system was particularly beneficial for those most in need of it. A similar pattern was observed in an earlier study by Macedonia and colleagues ([59]), which demonstrated that low-performing learners gained more from using semantically related gestures to support vocabulary retention than their high-performing peers.

Together, these studies extend the theoretical foundation of embodied cognition by showing that gesture-based learning is not the limit of motor involvement in language acquisition. When learners physically interact with objects, whether real or virtual, they engage perceptual and motor systems in a way that reinforces word meaning more directly and concretely. This line of research opens promising avenues for applying immersive, interactive strategies in language education, particularly through the use of virtual reality and other digital tools that support full-body engagement in learning.

## 7. Cognitive Mechanisms Underlying Gesture-Enhanced Vocabulary Learning

A variety of cognitive mechanisms have been proposed to explain why gestures are so effective in supporting second language vocabulary acquisition. These mechanisms converge on the idea that gestures do not merely accompany verbal input but actively shape how language is encoded, stored, and retrieved in the mind.

One foundational explanation comes from Allan Paivio’s dual coding theory ([81]), which posits that information is processed through two independent but interconnected systems: a verbal system and a nonverbal, imagery-based system. When learners observe or perform gestures while acquiring new vocabulary, they engage both systems simultaneously. The gesture offers a visual and motoric representation of the word, which complements the verbal input and establishes multiple retrieval cues. This dual pathway significantly increases the likelihood of successful recall, particularly in contexts requiring active retrieval.

Complementing this is the levels of processing framework developed by [18] ([18]), which asserts that deeper, more elaborative processing of information leads to more durable memory traces. Gesture production, especially when learners generate gestures themselves, requires active engagement with word meaning, not just passive repetition. This deeper semantic elaboration enhances memory consolidation, as learners must map physical movements onto lexical concepts in a meaningful way. For example, enacting a word like “to stir” not only illustrates the concept visually but also requires an internalized understanding of the action, reinforcing the lexical–semantic link.

Another mechanism that supports the efficacy of gestures is cognitive offloading, as described by Cook, Mitchell, and Goldin-Meadow ([17]). This concept suggests that gestures can reduce the cognitive load on working memory by externalizing parts of the learning process. By offloading elements of spatial or conceptual processing into the motor system, gestures free up cognitive resources that can then be devoted to encoding, organizing, and integrating new vocabulary. This is particularly relevant in L2 learning, where cognitive resources are often stretched due to the added demands of unfamiliar grammar, phonology, and orthography.

Related to this is the idea of embodied traces, a mechanism rooted in embodied cognition theories. According to [89] ([89]), gestures create motor memory traces that are tightly integrated with the corresponding linguistic representations. These sensorimotor traces can later be reactivated during recall, providing an additional access point for retrieving the target word. For instance, the motor pattern associated with a gesture may involuntarily re-emerge when trying to remember a word, subtly cueing the learner and facilitating retrieval. Neuroimaging studies have shown that gesture-based learning activates motor and premotor cortices, lending support to the view that linguistic knowledge becomes grounded in the body through gesture ([60]).

Finally, there is increasing recognition of the role of emotional engagement in vocabulary learning. Research by Macedonia and colleagues ([54]) suggests that the embodied, physical nature of gesture enhances emotional involvement with the learning material. When learners are actively moving and engaging with language through their bodies, the learning experience becomes more personal and emotionally salient. Since emotional arousal is known to facilitate memory encoding and retention, this heightened engagement likely contributes to the superior performance observed in gesture-enhanced learning conditions.

A recent review by [42] ([42]) examines how co-speech hand gestures contribute to emotional expression in communication. The findings suggest that gestures not only complement verbal messages but also enhance the emotional impact of communication, which can lead to improved engagement and memory in language learning contexts.

Together, these mechanisms offer a multi-dimensional account of why gestures are effective in second language vocabulary learning. They provide verbal–nonverbal redundancy, deepen semantic engagement, reduce working memory load, establish sensorimotor associations, enhance attention, and increase emotional salience, all of which are known to support robust and lasting learning.

## 8. The Body Builds Memory: Long-Term Effects

While gestures have been shown to enhance immediate vocabulary acquisition, a growing body of research suggests that their benefits extend well beyond the short term. One of the most compelling demonstrations of this comes from a longitudinal study conducted by [54] ([54]), in which participants were taught foreign language vocabulary using gesture-based techniques. Remarkably, even after a period of 14 months, with no experimentally conducted review or rehearsal, learners who had encoded vocabulary through gestures retained significantly more words than those who learned through traditional verbal methods alone. These findings provide strong evidence that gestural learning leaves a lasting imprint on memory, indicating that the physical enactment of word meanings creates robust, long-lasting memory traces.

The persistence of these effects points to a deeper involvement of the body in memory consolidation processes, suggesting that gestures are not merely beneficial at the moment of learning, but play an active role in how information is stored and stabilized over time. One explanation lies in the relationship between gesture-based learning and sleep-dependent memory consolidation. In addition, gesture-enhanced learning activates multiple, distributed neural systems across the brain ([60]). Unlike verbal learning alone, which tends to rely heavily on the language-related regions of the left hemisphere, gesture-based instruction recruits motor, visual, and sensory areas as well. According to [75] ([75]), memories that are distributed across several neural networks, especially those that include perceptual and action-related components, are more resilient to forgetting and interference. This multimodal encoding process provides redundancy, making it more likely that at least one retrieval path will remain accessible over time.

Another important factor is the involvement of procedural memory systems in gesture-based learning ([58]). While traditional vocabulary learning is typically confined to declarative memory, which is responsible for storing facts and events, gestural learning engages procedural memory, which governs skills and motor sequences. As [106] ([106]) has argued, procedural memory is less susceptible to rapid decay and tends to support more automatic, durable forms of knowledge. When vocabulary is learned through gesture, it may thus become partially embedded in procedural memory structures, giving rise to more stable, embodied lexical representations.

Moreover, gestures can facilitate memory not just during the encoding phase but also during retrieval. The motor component of gestures often acts as a self-generated retrieval cue. Even when learners are not explicitly instructed to repeat a gesture during recall, the original bodily action may be involuntarily reactivated, helping to trigger access to the associated word. This process resembles the effects of retrieval practice, widely known to strengthen memory, suggesting that gestures may serve as a kind of embodied retrieval support ([16]).

Taken together, these findings suggest that the body does more than support initial vocabulary learning; it actively contributes to the stabilization and durability of language knowledge over time.

## 9. Into the Brain: Neuroscientific Insights

Advances in cognitive neuroscience have deepened our understanding of why gesture-enhanced vocabulary learning yields such significant and lasting effects. Brain imaging studies, in particular, have illuminated the profound neural mechanisms underlying this multimodal learning approach. When learners acquire new vocabulary through gestures, the brain’s engagement goes far beyond the traditional language-processing areas. Research using functional magnetic resonance imaging (fMRI) has consistently shown that gesture-based learning recruits not only language-related regions such as the left inferior frontal gyrus and temporal cortices but also sensorimotor systems, including the primary motor cortex, the premotor cortex, and supplementary motor areas ([60]). Among others [66] ([66]) found that words learned with associated gestures elicited co-activation in both linguistic and motor regions, forming a distributed and integrated neural network that supports robust word retrieval.

These results suggest that gestures help encode vocabulary in a richer, more interconnected neural architecture. Unlike words learned through auditory or visual input alone, gesture-learned vocabulary is anchored in motor memory pathways, creating multimodal representations that are more resilient to forgetting. This distributed activation across cortical areas, encompassing language, movement, and perceptual processing systems, appears to facilitate not just encoding but also the stabilization and retrieval of lexical items.

The involvement of motor systems in language learning is further supported by studies using transcranial magnetic stimulation (TMS). In a demonstration of causality, [65] ([65]) applied TMS to temporarily disrupt activity in the motor cortex and observed that participants’ ability to recall gesture-learned vocabulary was significantly impaired. This result provides strong evidence that motor cortex activation is not merely a byproduct of gesture use but plays a functional and essential role in memory retrieval.

Further insights into the neural mechanisms underlying gesture-based vocabulary learning have come from neuroimaging studies examining how motor and language systems are co-activated during learning. In an fMRI study, [60] ([60]) found that foreign vocabulary learned with accompanying iconic gestures elicited activation in premotor areas. These findings suggest that iconic gestures engage motor-related brain regions in ways that reinforce semantic encoding, providing a neural basis for the observed memory benefits. This motor involvement probably reflects the creation of sensorimotor associations that deepen conceptual processing of the words.

In a related study, [45] ([45]) explored how gestures influenced the implicit retrieval of newly learned words. Their findings indicate that gestures facilitate access to word meaning during recall, supporting the idea that action-based representations can strengthen lexical memory. Although that study did not directly examine functional connectivity between motor and language networks, the behavioral outcomes are consistent with theories proposing an integrated role of the motor system in language processing.

In addition, areas of the brain responsible for multimodal integration, such as the angular gyrus ([10]; [108]) and posterior superior temporal sulcus ([95]), show heightened activity when learners retrieve vocabulary learned through gestures. [94] ([94]) argues that these regions serve as hubs where linguistic, motoric, and perceptual information are synthesized into unified memory traces. This binding of verbal and bodily representations probably contributes to the depth and durability of gesture-based learning.

Another feature of this process is the phenomenon of motor resonance. Pulvermüller and colleagues ([85]) demonstrated that simply retrieving words previously learned with gestures triggers activity in motor circuits. This automatic motor simulation indicates that the brain encodes and stores language not as an abstract symbol system alone but as something fundamentally grounded in bodily experience ([84]). Such embodied memory traces allow learners to involuntarily re-engage the original motor pattern during recall, thereby reinforcing the word’s meaning and thereby increasing the likelihood of successful retrieval.

Further evidence comes from studies on the somatotopic representation of action words. [33] ([33]) demonstrated that reading action-related words (e.g., “kick,” “pick”) activated specific regions in the motor and premotor cortex corresponding to the body parts involved in those actions. This somatotopic activation implies that the motor system is engaged during the processing of action-related language, supporting the notion of embodied semantics.

Taken together, this neuroscientific research paints a compelling picture of gesture-enhanced vocabulary learning as a deeply embodied, multisystem process. By engaging language, motor, perceptual, and integrative brain systems simultaneously, gesture use transforms vocabulary acquisition into a full-bodied cognitive experience. The brain does not treat gesture as a peripheral aid to language but incorporates it as a central component of the learning process. This integration results in more durable, retrievable, and richly encoded lexical memories ([44]), highlighting the neurological basis for the enduring success of gesture-based language instruction.

Beyond the spatial insights provided by neuroimaging techniques like fMRI, electrophysiological methods such as event-related potential (ERP) and magnetoencephalography (MEG) have offered valuable temporal perspectives on how gestures influence the brain’s processing of second language vocabulary. These methodologies allow researchers to examine the timing of neural responses with millisecond precision, providing a dynamic picture of how gesture-enhanced learning shapes language processing in real time.

One of the most consistent findings in this area involves the N400 component, a well-established marker of semantic processing. In studies where learners were taught new vocabulary using congruent gestures, the N400 amplitude elicited by these words during later recognition or comprehension tasks was significantly reduced. This reduction, observed in the work of [41] ([41]), is typically interpreted as evidence of facilitated semantic integration. In essence, gesture-enriched vocabulary is processed more efficiently at the meaning level, probably because the motoric and semantic associations formed during learning provide richer, more accessible representations of words’ meanings.

In addition to semantic facilitation, gestures also appear to influence syntactic processing, as indicated by shifts in the P600 component. [34] ([34]) found that vocabulary learned through gesture elicited earlier and more pronounced P600 responses during sentence comprehension. The P600 is generally associated with syntactic reanalysis or repair, and earlier onset suggests that gesture-learned words are more seamlessly incorporated into syntactic structures. This suggests that the benefits of gesture-based learning go beyond simple vocabulary retrieval, facilitating the seamless integration of words into broader grammatical structures. This finding is supported by [55] ([55]), who showed that words learned with gestures are more readily retrieved and more accurately used in producing sentences. Another electrophysiological signature of gesture involvement is the suppression of the mu rhythm, an EEG marker linked to motor system activity. Suppression of this rhythm, typically observed over sensorimotor cortices, is generally taken as evidence that motor circuits are being activated. In a study by [86] ([86]), participants were trained to manipulate objects. EEG recordings showed that during the observation of these movements, there was significant mu rhythm suppression over sensorimotor areas, indicating that the motor representations formed during learning were reactivated during observation. This supports the theory of embodied memory traces, where sensorimotor experiences enhance the encoding and retrieval of verbal information.

MEG studies have begun to uncover the oscillatory mechanisms that may underpin the effectiveness of gesture-enhanced language learning. Among these, gamma-band oscillations have attracted particular attention for their role in integrating multimodal linguistic information. Research by [22] ([22]) demonstrated that gamma-band synchronization occurs among key cortical regions such as the left inferior frontal gyrus and medial frontal areas during tasks involving covert verb generation. This synchronization, modulated by slower theta rhythms, suggests that gamma activity plays a central role in binding distributed neural systems involved in expressive language functions. Although the study did not focus directly on gesture learning, its findings provide a model for how linguistic and motor systems might be functionally connected through oscillatory mechanisms.

Supporting this, [23] ([23]) investigated how gestures influence neural oscillations during speech processing. Their results showed that the presence of meaningful gestures alongside degraded speech increased gamma-band power in the left superior temporal gyrus, an area closely associated with auditory and semantic processing. This enhancement was interpreted as reflecting greater neural integration of visual (gestural) and auditory (spoken) signals, enabling more efficient semantic processing under challenging listening conditions. These findings reinforce the view that gamma-band activity supports the integration of multimodal cues, including gesture and speech, during language comprehension.

Together, these electrophysiological findings underscore the intricate interplay between language and action systems in the brain. They show that gesture-enhanced vocabulary learning leaves measurable traces not only in terms of spatial brain activation but also in the fine-grained timing and coordination of neural events. From early semantic access to later syntactic integration and motor resonance, the entire timeline of word processing is shaped by prior embodied experience. Event-related potential (ERP) and magnetoencephalography (MEG) studies have provided temporal insights into gesture-enhanced vocabulary processing. These electrophysiological markers provide fine-grained temporal evidence of how gesture-enhanced learning affects neural processing at different stages.

Causal evidence for the involvement of motor systems in gesture-enhanced vocabulary learning has been substantiated through various neuroscientific intervention studies. One notable approach involves transcranial magnetic stimulation (TMS). In a study by [65] ([65]), participants learned foreign language vocabulary accompanied by either gestures (sensorimotor enrichment) or pictures (sensory enrichment). Following training, TMS was applied to the primary motor cortex during a translation task. The results indicated that TMS disruption selectively impaired the translation of gesture-enriched words, while leaving picture-enriched word translation unaffected. This finding suggests a causal role of the motor cortex in processing vocabulary learned through sensorimotor experiences.

Collectively, these studies provide compelling evidence that the motor system is not merely associated with but is causally involved in the representation and retrieval of vocabulary learned through gestures.

## 10. Age-Specific Effects of Gesture-Based Vocabulary Instruction

Gesture-enhanced vocabulary instruction demonstrates varying effectiveness across the human lifespan, shaped by developmental changes in cognition, motor skills, and brain maturation. Understanding these developmental trajectories is essential for designing pedagogical strategies aligned with learners’ evolving capacities.

In early childhood (ages 3–6), children benefit substantially from embodied learning, especially gesture-based instruction. Their natural inclination toward sensorimotor exploration, a key component of native language acquisition, makes them highly receptive to physically grounded input. Studies by [67] ([67]) and [17] ([17]) have shown that combining movement with linguistic input enhances memory retention.

During middle childhood (ages 7–11), refined motor skills and improved executive functions, such as attention regulation and working memory, allow children to better coordinate physical and verbal inputs. Research by [101] ([101]) and [2] ([2]) confirms that children in this stage show robust gains in vocabulary learning and retention through gesture-enriched methods, with effects persisting over months. This developmental window is especially sensitive to embodied approaches, as cognitive and motor systems interact more effectively under guided instruction.

In adolescence (ages 12–17), gesture-based learning remains highly effective and may even intensify in impact. [63] ([63]) observed that adolescents benefit more from gesture-based instruction as they age, likely due to the maturation of frontal-motor neural pathways and the emergence of metacognitive awareness. These learners become more capable of using gestures deliberately as mnemonic tools and of adapting them to abstract concepts. The deepening integration of sensory, motor, and cognitive systems during this period enhances the encoding and retrieval of vocabulary through embodied strategies.

Among young adults (ages 18–30), who typically exhibit peak cognitive and motor functioning, gesture-enhanced learning is both effective and well-documented. Most studies cited in this paper demonstrate improved retention for both concrete and abstract vocabulary when gestures accompany learning. However, individual differences such as learning preferences, working memory capacity, and educational background moderate the benefits. For most learners, gestures serve as a powerful supplement to verbal instruction by strengthening multimodal encoding and recall pathways.

Although research in older adults (60+) has been less extensive, emerging evidence suggests that gesture-based learning offers substantial benefits. As age-related decline affects episodic and verbal memory, procedural and motor memory remain relatively preserved ([105]). Studies by [76] ([76]), [14] ([14]), and [93] ([93]) demonstrate that self-enactment and visual gestures support vocabulary retention and speech comprehension, particularly under degraded listening conditions.

These age-specific patterns are rooted in neurodevelopmental mechanisms. Myelination and synaptic pruning enhance the efficiency of neural circuits connecting language and motor regions during childhood and adolescence ([39]; [47]). Executive functions and metacognitive abilities mature steadily, enabling learners to use gestures more strategically ([20]; [50]). Older learners benefit from extensive semantic networks that serve as scaffolds for new vocabulary, supporting integration through multimodal associations ([38]).

Altogether, gesture-based vocabulary learning evolves dynamically across the lifespan, yet its fundamental advantages remain consistent. From early childhood through late adulthood, gestures offer a neuroscientifically grounded, adaptable tool for enhancing language learning. These findings highlight the importance of age-responsive instructional design that aligns with learners’ developmental profiles to fully leverage the cognitive and neural strengths present at each life stage.

## 11. From Theory to Classroom: Instructional Design Principles

Several instructional design principles emerge from this research base. Foremost among them is semantic congruence ([35]). Gestures should meaningfully reflect the target word’s semantics, with iconic gestures for concrete terms and metaphoric gestures for abstract ones, as gestures that match a word’s meaning enhance memory through dual coding and embodied grounding ([31]; [40]; [60]; [101]).

Active production of gestures yields stronger memory effects than passive observation ([30]). Learners who physically perform gestures remember vocabulary better than those who merely watch them ([62]; [69]). This aligns with theories of motor trace activation and procedural memory involvement ([107]).

Timing also matters. Gestures should be performed simultaneously with verbal input to foster optimal multimodal integration. [52] ([52]) showed that synchronizing movement with speech activates both language and motor areas of the brain, thereby reinforcing cross-modal associations that strengthen memory encoding.

Practice structure further modulates retention. [89] ([89]) found that distributed practice, where gesture-word pairs are reviewed across multiple sessions, leads to more durable learning than massed repetition. Rehearsing gesture-linked vocabulary over spaced intervals enhances consolidation and reduces forgetting.

Sequencing instruction from concrete to abstract vocabulary can also scaffold learners more effectively. That is to say that starting with words that lend themselves to simple, easily performed gestures helps build confidence and supports the transition to more complex lexical items ([55]; [72]; [77]).

Contextual embedding is equally important. Gesture-vocabulary learning is most effective when embedded in meaningful, communicative tasks rather than taught in isolation. Situating gestures within narratives, dialogues, or real-life scenarios allows learners to transfer their embodied knowledge to functional language ([1]; [52]).

Lastly, personalization plays a crucial role in engagement and memory. Allowing learners to tailor or invent their own gestures makes the learning process more personally relevant. [67] ([67]) found that such learner-driven gesture creation enhanced both motivation and recall in early childhood learners, and similar benefits have been observed in adolescent and adult populations.

Together, these principles offer a framework for the design of gesture-enhanced language instruction. They emphasize the importance of intentionality, alignment with cognitive mechanisms, and responsiveness to learner characteristics and contexts. By grounding classroom practice in empirical research, educators can harness the full potential of gesture to support lasting and meaningful vocabulary acquisition.

## 12. Conclusions

This paper argues for a reconceptualization of second language vocabulary acquisition by foregrounding the body as an active cognitive tool. Drawing from a wide range of empirical research in psychology, neuroscience, linguistics, and education, this paper shows that gesture and sensorimotor engagement are not peripheral but central to how words are encoded, retained, and retrieved in long-term memory. The prevailing legacy of Cartesian dualism in educational practice, where learning is conceptualized as a purely mental activity, stands in contrast to the converging evidence supporting embodied cognition. Language is not acquired in abstraction; it is grounded in physical experience, enacted through the body, and shaped by our interactions with the world.

Empirical studies have consistently demonstrated that learning vocabulary with semantically congruent gestures enhances both immediate and delayed recall more effectively than traditional, disembodied methods. Iconic gestures, enactments, and even arbitrary but consistent movements engage the motor system in ways that deepen semantic processing and create durable memory traces. These benefits are not limited to any one age group; rather, they evolve developmentally, offering age-appropriate advantages across the lifespan, from preschoolers to older adults. Furthermore, neuroscientific investigations have shown that gesture-based learning activates a distributed network of brain regions, including premotor, motor, and multimodal integration areas, confirming that embodied learning recruits systems far beyond those typically associated with language.

More recent studies extend this research into digital and virtual environments, showing that even virtual agents and graspable virtual objects can replicate the cognitive benefits of physical gesture. This opens promising pathways for designing intelligent, embodied learning technologies that can adapt to individual learner profiles, particularly benefitting those with lower aptitude or specific learning challenges.

Theoretical frameworks such as embodied and grounded cognition, perceptual symbol systems, and connectionist models support these findings by positing that language is intrinsically tied to the body and sensorimotor systems. Thus, language learning is not a mere symbolic decoding exercise but a deeply interactive, multimodal process intrinsically tied to the body and thus rooted in action, perception, and emotion. As such, gestures are more than communicative supplements or mnemonic devices; they are mechanisms of thought that help transform unfamiliar vocabulary into personally meaningful, experientially grounded knowledge.

Pedagogically, this body of evidence calls for a shift in how vocabulary is taught in second language classrooms. Instructional design should intentionally incorporate semantically congruent, actively produced gestures synchronized with verbal input. Practices should be distributed over time, embedded in meaningful contexts, and, where possible, tailored to learners’ developmental stages and individual preferences. The body, long excluded from formal learning environments, must be reinstated as a co-constructor of knowledge.

In sum, this paper has shown that using the body, through gesture and action, is not merely helpful but essential for optimizing second language vocabulary learning. By aligning instructional practices with how the brain and body naturally encode meaning, educators can unlock new levels of engagement, retention, and fluency. The future of language learning lies not only in our heads but also in our hands.

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
