# Peer review of "Your Body as a Tool to Learn Second Language Vocabulary"

_behavsci, 2025, doi:10.3390/bs15080997_

Round 1

Reviewer 1 Report

Comments and Suggestions for Authors

A very well written manuscript which I really enjoyed reading. A couple of comments from me as a reviewer. I thought that it integrates perspectives from linguistics, cognitive science, neuroscience, and pedagogy and builds a strong argument for gesture-based learning in second language (L2) acquisition. I also thought the manuscript was well-referenced using scholars such as Wilson, Barsalou, and Di Paolo. This manuscript showed a strong theoretical framework. I also think that the works of embodied cognition was well documented by including recent research to support your work. One of the manuscript's strengths is the organization including theories and results from other studies. The manuscript ended strongly by encouraging the conversation beyond gestures for language teaching and the possibility of using technology. However, there were some sections, especially the behaviorial and neuroscience parts that had similiar studies. The manuscript could have showed a more balanced view by including challenges of learning second language in the classroom as well as assessing the methodological limitations from other studies. A couple of scientific terms could be defined and introduced to the readers more clearly. More information on how the technology could advance second language learning in the manuscript could be expanded. Overall, a very nice work and I would recommend to accept with minor revisions.

Comments on the Quality of English Language

Below are a few sentences that I would suggest for revising:

Line 28-30: Hulstijn (2001) highlights the limitations of decontextualized learning, in which vocabulary items are presented in isolation, lacking the meaningful contexts that support deeper cognitive processing and long-term retention. 

Line 111 – Delete “In her influential paper”….seems subjective.

Lines 175-177: Contrary to the traditional view that cognition relies on amodal, abstract symbols detached from sensory experience, Barsalou (2008) proposed that cognitive processes, including language, are fundamentally grounded in perceptual and sensorimotor experiences.

Line 204 – fix the “about” quote

Author Response

The author would like to thank the Reviewer for their thoughtful and encouraging comments. The author greatly appreciates the recognition of the manuscript's theoretical integration and organization, as well as the constructive suggestions for improvement. Please find below my detailed responses to the specific points raised:

Comment: There were some sections, especially the behavioral and neuroscience parts, that had similar studies. The manuscript could have showed a more balanced view by including challenges of learning a second language in the classroom, as well as assessing the methodological limitations from other studies.

Response:
The author appreciates this suggestion. Regarding the challenges of second language learning in classroom settings, the  agrees that this would be an interesting and valuable angle to explore. However, this lies outside the scope of the author's current line of research, and she does not have the breadth of expertise to confidently review or take a critical position on this area.

As for assessing methodological limitations, the had addressed this in the manuscript, specifically in lines 77–84, where she had discussed some of the weaknesses and constraints in prior studies.

Comment: A couple of scientific terms could be defined and introduced to the readers more clearly.

Response:
Thank you for this remark. The author would kindly ask the reviewer to specify which terms they feel require clearer definition, so that I can revise the manuscript accordingly and ensure accessibility for a broader readership.

Comment: More information on how technology could advance second language learning in the manuscript could be expanded.

Response:
The Reviewer raises an excellent point. In fact, the author was tempted to elaborate on this topic. However, she ultimately decided against doing so to maintain a focused and concise manuscript, as the inclusion of this discussion would have significantly expanded its length. The author agrees that the potential for technology to support embodied approaches to second language (vocabulary) learning is both timely and important. For this reason, the author plans to address this topic more thoroughly in a separate paper in the near future.

Comment: Comments on the quality of English language – suggested sentence revisions

  • Line 28–30: Hulstijn (2001) highlights the limitations of decontextualized learning, in which vocabulary items are presented in isolation, lacking the meaningful contexts that support deeper cognitive processing and long-term retention.
    Revised as suggested.

  • Line 111: Delete “In her influential paper” (subjective).
    Done.

  • Lines 175–177: Contrary to the traditional view that cognition relies on amodal, abstract symbols detached from sensory experience, Barsalou (2008) proposed that cognitive processes, including language, are fundamentally grounded in perceptual and sensorimotor experiences.
    Revised as suggested.

  • Line 204: Fix the “about” quote.
     Corrected.

Once again, the author sincerely thanks the reviewer for their generous and insightful feedback. These suggestions have helped strengthen the manuscript.

Reviewer 2 Report

Comments and Suggestions for Authors

This is an outstanding, much needed review article for all researchers, reflective practitioners and course designers that reminds us all that language learning– very much like ordinary language practice– is not a mere "symbolic decoding exercise" but "a deeply interactive, multimodal process" which is "intrinsically tied to the body and sensorimotor systems" (911-914). Strangely enough, ordinary language teaching and learning strategies disregard the reality of the embodied cognitive mechanisms at play in language production and reception. The authors take stock of the powerful intellectual, empirical and neurological arguments that clearly establish that "gestural learning" (i) is consistent with what we know of genreral brain functioning and embodied cognition (ii) has a positive impact on comprehension and retention. They rightly claim that limiting the contribution of gesture (and more generally movement) to iconic (re)enactments of meanings, is only part of the story, that metaphoric gesticulation helps (everyone) with abstraction and that the interplay between "sensorimotor engagement"  and gesture symbolism can lead to the creation and "manipulation of virtual objects".

As a gesture scholar myself and a language teacher, I wholeheartedly embrace the idea of "grasping-based learning". I would personally  add a few references to general semiotics (e.g. Ecco, Kendon but also McNeill and Calbris) because none of this would work were it not for the  general symbolic properties of space, objects (concrete > abstract)  and bodily motion. As Kendon rightly claims in his treatise on gesture, gesture symbolism is symbolism in action, and meaning-making can be construed as "fabricating" activity that crucially (although not exclusively) relies on the hands. This claim has been taken up by Jürgen Streeck. As things stand, the authors might give the wrong impression that they are in denial of symbolic activity, when they are not. They should consider adding "mere" (line 913) ("not a MERE symbolic decoding exercise"). This article scans all the powerful arguments in favor of embodied, action-based  approaches to language teaching and learning. I have myself extensively written on the interest of "engaging the learning body in language education" (Lapaire 2019) but without providing all the scholarly references and detailed arguments that the authors usefully supply. I think that what they say of vocabulary acquisition is likely to be extendable to grammar as well, that grammatical meanings and processes lend themselves to gestural strategies of deconstruction, interpretation and memorization too. And since the authors rightly remind us that words normally occur in context  and are best learnt that way, I would be interested to know how they would envisage the integration of word-specific gestures within broader, gesture-supported, syntactic constructions. My experience is that gestures differ in form and function, when the focus is strictly on de-alienating foreign words and when gestures help with the delivery of full sentences. In the latter case, different moves and postural adjustments function as mere differentiators (differential markers), not explainers. And they work very well too. 

I am sincerely looking forward to the publication of this very nicely written and thought out piece of reflection. I am impatient to quote it and share when it is released.

Author Response

I would like to sincerely thank Reviewer 2 for their enthusiastic and generous comments. It is deeply encouraging to receive such positive feedback from a fellow gesture scholar and language teacher. I truly appreciate the thoughtful suggestions and the constructive dialogue opened by this review. Below are my detailed responses:

Comment: “This is an outstanding, much needed review article… reminds us that language learning is not a mere ‘symbolic decoding exercise’ but a deeply interactive, multimodal process… tied to the body and sensorimotor systems.”
Acknowledged and appreciated. The phrasing “not a mere symbolic decoding exercise” has been adopted (line 913) to better reflect the nuanced nature of the critique.

Comment: “Strangely enough, ordinary language teaching and learning strategies disregard the reality of embodied cognitive mechanisms... gestural learning is consistent with general brain functioning and has positive impact.”
Acknowledged. This observation is central to the manuscript’s position, and I appreciate the Reviewer’s reinforcement of this point.

Comment: Consider adding references to general semiotics (e.g., Eco, Kendon, McNeill, Calbris), as these symbolic properties underpin gestural communication.
Addressed. McNeill is cited explicitly in lines 235–240, and a dedicated section on Kendon’s semiotic perspective on gesture—as “symbolism in action”—has been added. A section on Streeck’s contribution has also been included, expanding on his concept of “conceptual gestures” and their emergence through repeated, embodied interaction. The symbolic properties of space and motion are now more explicitly acknowledged in the theoretical framing.

Comment: “As Kendon rightly claims…” — the author has added a section on Kendon’s work.
 Done. A detailed paragraph summarizes Kendon’s 2004 treatise Gesture: Visible Action as Utterance, emphasizing gesture as a semiotic process and embodied utterance.

Comment: “This claim has been taken up by Jürgen Streeck…” — I have added a section on Streeck’s work.
Done. A new paragraph presents Streeck’s views on the origin and social sedimentation of gesture, highlighting how hands-on activity transforms into symbolic communication. This section clarifies the author’s recognition of symbolic dynamics within embodied interaction.

Comment: “As things stand, the authors might give the wrong impression that they are in denial of symbolic activity.”
Clarified. The use of the word “mere” in line 913 now reflects that language is not only symbolic but is fundamentally embodied and interactive, acknowledging both dimensions.

Comment: Reference to Lapaire (2019) added and discussed.
Done. A summary paragraph has been added, presenting the main ideas of Engaging the Learning Body and recognizing Lapaire’s influence on the manuscript’s theoretical direction.

Comment: Gesture-based strategies may extend to grammar as well as vocabulary.
Acknowledged. I fully fully agrees with this point. In an earlier book for practitioners, I described the use of gesture to teach syntax and morphology.  I will be happy to provide the reference after the review process.

Comment: Interest in how gesture might function differently for isolated vocabulary versus syntactic constructions.
Acknowledged and appreciated. This observation accurately reflects my teaching experience. When working with full sentences, gestures often serve more as differential or prosodic markers than as explicit semantic carriers. I would be delighted to continue this conversation and would very much welcome a Zoom meeting once the review process is complete.

Comment: “I am sincerely looking forward to the publication… I am impatient to quote it.”
Your words mean a great deal, and I am honored by your interest in the work. Thank you so much!